# First Report of an Asymptomatic *Leishmania (Viannia) shawi* Infection Using a Nasal Swab in Amazon, Brazil

**DOI:** 10.3390/ijerph19106346

**Published:** 2022-05-23

**Authors:** Luciana P. Oliveira, Luciana C. S. Nascimento, Fabiola S. Santos, Jaqueline L. C. Takamatsu, Luiz R. P. Sanchez, Walter S. Santos, Lourdes M. Garcez

**Affiliations:** 1Ministério da Saúde, Secretaria de Vigilância em Saúde, Instituto Evandro Chagas, Ananindeua 67015-120, PA, Brazil; lcassiasn3@gmail.com (L.C.S.N.); fabiolasantos@iec.gov.br (F.S.S.); waltersantos@iec.gov.br (W.S.S.); lourdesgarcez@iec.gov.br (L.M.G.); 2Hospital Municipal de Tomé Açu, Tomé-Açu 68680-000, PA, Brazil; jaque.takamatsu@gmail.com (J.L.C.T.); pinarafael@hotmail.com (L.R.P.S.); 3Departamento de Patologia, Universidade do Estado do Pará, Belém 66050-540, PA, Brazil

**Keywords:** cutaneous leishmaniasis, epidemiology, cotton swab, etiology

## Abstract

The state of Pará has recorded seven *Leishmania* species that cause tegumentary leishmaniasis (TL). *Leishmania* species induce distinct immunological responses from the host and exhibit resistance to Glucantime, the first-line drug treatment for TL in Brazil. Objective: Identify the etiology of TL in an Amazonian city in the state of Pará. Material and methods: Eleven patients with TL were recruited and nasal swabs, lesion swabs, and skin fragments samples were collected. In the control group (n = 6), only the nasal swabs were collected. Polymerase Chain Reaction (PCR) amplification of the gene region *hsp70-234* was performed using the extracted DNA from the samples, from which nine patients with TL and five in the control group were positive. Products were sequenced, mounted in CAP3 software, aligned using MAFFT v.7.221, edited in Geneious software v.8.1.7, and compared and aligned with sequences available in GenBank using the BLAST tool. Results: For patients with TL, six molecular diagnosis at the species level (*L.* (*Viannia*) *braziliensis* (n = 5/9), *L.* (*Viannia*) *shawi* (n = 1/9)) and three at the genus level (*Leishmania* sp. (n = 3/9)) were obtained. In the control group, four individuals were infected with *Leishmania* sp. (n = 4/5) and *L. (V.) shawi* (n = 1/5). Conclusion: This is the first report of *L. (V.) shawi* infection in the mucosal secretion of a healthy person in Brazil. Moreover, genetic variants were identified in the haplotypes of *L. (V.) braziliensis* in the gene sequence *hsp70-234.*

## 1. Introduction

In the Brazilian Amazon, several species of *Leishmania* cause TL. The state of Pará, the second-largest extension in this region, is the only state in Brazil to record seven dermotropic species of *Leishmania* pathogenic to humans, presenting variable levels of pathogenicity and drug resistance [1,2].

Lesional pleomorphism in patients with TL results from a balance in the parasite–host relationship. *Leishmania* species induce different immunological responses in the host. Three important species are associated with the severe evolution of the disease: *Leishmania (Viannia) braziliensis*, *Leishmania (Viannia) guyanensis*, and *Leishmania (Leishmania) amazonensis* [3,4].

Some *Leishmania* species are more prone to hematogenous dissemination. *L. (V.) braziliensis* causes more frequently disseminated conditions, being the main species causing mucosal leishmaniasis [4]. *L. (**V.)* braziliensis can be detected in the nasal mucosa of both sick and healthy people living in areas with endemic TL [5]. *L. (V.) guyanensis,* which is predominant in certain endemic areas of northern Brazil, notably in the states of Amazonas and Amapá [6], can also cause mucosal lesions [7] and is highly resistant to Glucantime, the first-line drug treatment for TL in Brazil [2]. Meanwhile, *L. (L.) amazonensis* can induce severely disseminated disease and anergic diffuse leishmaniasis [8]. Other pathogenic species, including *Leishmania (Viannia) lainsoni*, *Leishmannia (Viannia) shawi, L. (Viannia) naiffi*, and *L. (Viannia) lindenbergi*, do not usually cause complications. *Leishmania (V.) lainsoni* and *L. (V.) naiffi* usually present as a single lesion. *L. (V.) lindenbergi* has localized cutaneous lesions. No case of the mucocutaneous disease has been reported in any of the species [9].

*L. (V.) shawi* was isolated for the first time from sloths and monkeys from the Serra dos Carajás, Pará, Brazil, which is one of the etiological agents of TL in Brazil, and is associated with the cutaneous form of TL, usually observed as a single lesion without mucous or other complications [9,10]. *L.* (*V.*) *shawi* has been identified in sandflies, *Lutzomyia (Nyssomyia)*
*whitmani* [11], and *Lutzomyia gomezi* [12]. There are reports of *L. (V.) shawi* in the Amazon region identified in TL carriers [2,13].

The molecular characterization of *Leishmania* sp. that causes TL in the Amazon is important not only to define a prognosis for the patient but also to guide the therapeutic approach. In addition, it is essential to evaluate the distribution of *Leishmania* sp. and the occurrence of variants, and alert surveillance services to support transmission prevention [6,14]. The great challenge in the Amazon is to provide access and resources of scientific advances in communities to improve molecular diagnostic techniques [15]. Molecular methods have received increasing attention due to the possibility of distinguishing species in material obtained directly from the lesion. The enzymatic techniques that are effective for species distinction depend on the isolation of *Leishmania*, which makes their use in routine diagnostics impossible. The PCR in its various versions and with different markers are able to discriminate *Leishmania* species, but the sequencing of products with certain markers is what confirms the species distinction. Among the most used markers in PCR are G6PhD, ITS1, and Hsp70-234 [16]. In the Brazilian Amazon region, it obtained excellent results using the marker Hsp70-234 [6,17]. The study aimed to identify the etiology of TL in an Amazonian city in the state of Pará. Despite the difficulties in performing the study during the COVID-19 pandemic, molecular analysis of several clinical samples revealed human infection by *L. (V.) shawi* in nasal swabs for the first time. Moreover, we discuss the importance of this finding on the epidemiological surveillance of TL.

## 2. Material and Method

### 2.1. Study Site

The research was conducted at the Municipal Hospital of Tomé-Açu. The city of Tomé-Açu is located in the northeastern region of the state of Pará, Brazilian Amazon, and is an important agricultural and mining center in the state (Figure 1) [18,19]. In the municipality of Tomé-Açu there are two rainy seasons a year, one from December to May, representing 80% of the rainfall, and another from June to November. The locality features the following general characteristics: average annual temperature of 26 °C and rainfall average of 2300 mm/year [19].

### 2.2. Participant Recruitment

Participants were recruited in two stages: the first stage from October 2019 to February 2020, and the second from August 2020 to October 2020. The stages chosen were related to the increase in cases during the rainy season, and also to changes in health service due to the established pandemic.

Men and women over 18 years of age with a confirmed diagnosis of TL and controls were included in the study. The controls were individuals without injury, companions of the patients with TL who agreed to participate in the research, or those who live with the patients.

### 2.3. Participants and Sample Types

For patients with TL (n = 11), nasal swab, lesion swab, and skin fragment samples were collected. In the control group (n = 6), only nasal swabs were collected. All participants gave their consent for inclusion in the study. The study was conducted according to the Declaration of Helsinki, and the protocol was approved by the Research Ethics Committee of the Instituto Evandro Chagas (Opinion number: 3.601.679).

### 2.4. Sample Collection

Sample collection was performed after site cleaning and antisepsis, in the following order: (1) nasal swab, (2) lesion swab, and (3) skin biopsy. For the nasal swab, the absorbent segment of the swab was inserted into the anterior third of the nose and rotated clockwise (360°) over the nasal mucosa. The same procedure was performed in both teins with the same swab. For the lesion swab, the absorbent segment was pressed parallel to the edge of the lesion and rotated on its surface clockwise [20,21]. Meanwhile, the skin sample was obtained using a disposable punch (5 mm) after local anesthesia (2% lidocaine). This was performed on the infiltrated and eysthematous edge of the lesion [14].

### 2.5. Sample Storage

The end rod of each swab containing the sample was cut with the aid of sterile scissors and placed in a 1.5 mL microtube containing 500 μL of NET (NaCl 0.15 mM; EDTA 50 mM; Tris-HCl 0.1 M, pH 7.5). Skin samples were also preserved in NET. All tubes containing NET samples were stored at 4–8 °C.

### 2.6. DNA Extraction

DNA extraction from the samples was performed using the Wizard^®^ Genomic DNA Purification Kit (Promega, Madison, WI, USA) according to the manufacturer’s recommendations, with a final volume of 25 μL.

### 2.7. Polymerase Chain Reaction (PCR) for Target Region Hsp70-234

The target region *hsp70-234* (234 bp) was amplified using conventional PCR. Briefly, the PCR mix, with a final volume of 50 μL, contained 0.03 U/μL Taq DNA polymerase, 1.5 mM MgCl_2_, 0.25 mM of each dNTP), 1X buffer solution with KCl, 0.25% DMSO, 0.2 pMol of primers F (5′-GGACGAGATCGAGCGCATGGT-3′) and R (5′-TCCTTCGACGCCTCCTGGTTG-3′), and 3.0 μL of DNA from samples. The PCR conditions were as follows: initial denaturation at 94 °C for 5 min, followed by 32 cycles of denaturation at 94 °C for 30 s, amplification at 61 °C for 1 min, and extension at 72 °C for 1 min; and a final extension at 72 °C for 10 min. Two positive controls containing the genomic DNA of *L. (L.) infantum* (CEPA 6445) and *L.* (*V.*) *braziliensis* (CEPA 2904), and a control without DNA (white) were used. The amplified samples electrophoresed in 2% agarose in TAE buffer (Tris- acetate 0.004 M; EDTA 0.001 M, pH 8.0) containing GelRed™ (Biotium^®^, Fremont, CA, USA) at a concentration of 0.5 μg/mL and subsequently observed in a transilluminator for qualitative analysis [16].

### 2.8. Phylogenetic Sequencing and Analysis

PCR-*hsp70-234* products were purified with the EXOSAP-ITTM Express PCR Product Cleanup (AppliedBioystems^®^, University Park, IL, USA) and prepared for sequencing using the BigDye Terminator Kit v3.1 (Thermo Fisher Scientific^®^, Waltham, MA, USA) according to the manufacturer’s instructions. The forward and reverse sequences were obtained using the ABI3500XL automatic DNA analyzer. The sequences obtained were mounted in CAP3 software (Iowa State University, Ames, IA, USA), aligned using MAFFT v.7.221 (Universitu of Osaka, Osaka, Japan) [22], edited in Geneious v.8.1.7 software (Biomatters Ltd., Auckland, New Zealand) [23], and compared with sequences available in GenBank using the BLAST tool. Phylogenetic analyses were performed in three stages: (i) determination of the evolutionary model and likelihood analysis using the IQ-TREE v.1.3.2; (ii) phylogenetic reconstruction in IQ-TREE v.1.3.2 software (Center for Integrative Bioinformatics Vienna, Wien, Austria) using a non-parametric reliability test of 1000 replicates and bootstraps; and (iii) editing of the phylogenetic tree using FigTree v.1.4.2 software (University of Edinburgh, Edinburgh, UK) [24].

## 3. Results

Among the 11 patients with TL included in the study, six received molecular diagnosis at the species level (*L.*
*(V.) braziliensis* (n = 5/9), *L.* (*V.*) *shawi* (n = 1/9)), and three at the genus level (*Leishmania* sp. (n = 3/9)). For two individuals, it was not possible to amplify the deoxyribonucleic acid (DNA) of the respective samples. Furthermore, the sensitivity of PCR to amplify *hsp70-234* varied according to the sample collection. The highest number of positive results was obtained from the biopsy samples (n = 7/11), followed by injury swabs (n = 6/11) and nasal swabs (n = 4/11). Species identification also occurred in a greater proportion of the biopsy samples (n = 6/7). In the group of healthy individuals (n = 6), five had *hsp70-234*-positive PCR results in nasal secretion swabs, where *L. (V.) shawi* (n = 1/5) and *Leishmania* sp. (n = 4/5) were identified (Table 1).

A healthy participant diagnosed with *L. (V.) shawi* (4S) in nasal secretion was a parent that was accompanied the individuals with TL (4D), who also had the infection confirmed by *L. (V.) shawi*.

Most patients were men (8/11). The eight individuals were infected by *L.* (*V.*) *braziliensis* (n = 4/8), *L.* (*V.*) *shawi* (n = 1/8), and *Leishmania* sp. (n = 3/8). The nine patients with a molecular diagnosis of TL had ulcerated lesions and mostly localized lesions in the lower limbs (n = 7/9). The size of the lesions varied between 2 cm and 32 cm, and smaller lesions (2–6 cm in diameter) were more frequent (n = 4/9). The sociodemographic and clinical data of the nine individuals, for whom molecular diagnosis was possible, are shown in Table 2.

Eight nucleotide sequences were obtained: six were identified as *L.* (*V*.) *braziliensis,* with four haplotypes; and two were *L.* (*V*.) *shawi*, corresponding to only one haplotype (Table 3).

The sequences of *L.* (*V.) braziliensis* from the study (MZ399588, MZ399590, MZ399591, MZ399592, MZ399594, and MZ399595) clustered with GenBank sequences (KY249628 and KY249630) that also originated in Tomé-Açu (Amazon Biome), with values between 90 and 100 bootstraps. KY249629, KY249631, and KY249632 did not align with the nucleotide sequences in the present study, indicating a probable mutation in the gene region *hsp70-234* of *L.* (*V.*) *braziliensis* circulating in the municipality. In the phylogenetic analysis, sequences MZ399591 and MZ399592, which belonged to the same patient, were grouped into a different clade from the other sequences of *L.* (*V.*) *braziliensis* (Figure 2).

The sequences of *L. (V.) shawi* identified in this study (MZ399593 and MZ399589) were grouped with nucleotide sequences of the same locality (GU071177.1 and GU071175.1) in a clade with values between 70 and 89 bootstraps, proving the discrimination of the species.

## 4. Discussion

A higher amplification rate of the gene region *hsp70-234* was observed in DNA samples from skin biopsy, which also favored the discrimination of *Leishmania* species using sequence analysis. Swab is a collection method considered useful for public health because it is painless and easy to perform and can be performed by trained professionals of the health surveillance team and not necessarily physicians. Since it is not invasive, swab sampling offers more comfort and less risks, and is important for health surveillance in places with structural difficulties [21,25,26,27,28]. Moreover, the efficacy of swab compared to biopsy can be used for the discrimination of *Leishmania* species, and swab could be indicated as an aid in health services, in support of the diagnosis of the disease, especially when the parasitological examination of the exudate obtained through scarification of the lesion (blade-colored smear) is negative [29].

In this study, *Leishmania* DNA was found in nasal secretions, including the first report of *L.* (*V.) shawi* in the nasal mucosa of a healthy woman, accompanied by a male patient infected with *L. (V.) shawi.* The species was identified in patients with TL at the study site [2], and in other locations in the Brazilian Amazon, in the city of Roraima [13]. In the municipality of Tomé-Açu, the sandfly species are present in extradomiciliary environments [18].

*Leishmania* RNA virus 1 (*LRV1*) was found in patients diagnosed with TL due to *L.* (*V.) shawi* infection in the Brazilian Amazon [30]. *Leishmania* infection by *LRV1*, a double-taped RNA virus of the Totiviridae family, is associated with increased mucosal injury [29]. There are strong indications that viruses infecting *Leishmania* spp. can increase the survival of the parasite in humans and influence the pathogenicity of the disease, as well as the drug resistance of the parasite [31,32]. The frequency of this virus is twice as high in mucosal lesions than in cutaneous lesions [32,33]. *LRV1* mainly infects *L.* (*V.*) *braziliensis* and *L.* (*V.*) *guyanensis*, both of which are capable of producing mucous lesions and are resistant to the first-line of treatment for TL—Glucantime^®^ [2]. These two species were reported in the same area as in the present study. We previously identified cases of human CL caused by *L.* (*V.) shawi* [1].

The findings of this study provide a warning for the possible association between *L.* (*V.*) *shawi* and mucous lesions, since *L.* (*V.*) *braziliensis* can be found both in normal mucosa and in injured mucosa [5]. Experimental studies showed that infection by *L.* (*V.) shawi* without symptoms in mice can occur through the activation of antigens capable of inducing a protective immune profile, which is important for the development of vaccines against TL [34,35]. These observations allow for further studies in the fields of genomics and bioinformatics in this species of *Leishmania*.

The nucleotide sequences of *L. (V.) shawi* identified in this study (MZ399593 and MZ399589) corresponded to haplotype V and were grouped in the same clade as other sequences obtained from GenBank, suggesting that there was no genetic variation in the sample group.

The presence of the parasite in individuals without active disease can be observed in endemic areas, especially in the blood samples, nasal mucosa smears, conjunctiva, or tonsils. The presence of *Leishmania* in mucous tissues without signs and symptoms may also be considered a common feature of the natural history of infection by *Leishmania* species of the subgenus *Viannia* [36]. Studies on the role of asymptomatic *Leishmania* carriers in the transmission of leishmaniasis are limited, and whether community and population control interfere with the transmission requires further understanding [37].

*L.* (*V.) braziliensis* was the most frequent species in the individuals and was also found in the mucosal secretion of patients, but only *L.* (*V.) shawi* was detected both in a male patient (skin lesion) and in a healthy individual (nasal swab). We have previously identified a high frequency of *L.* (*V.*) *braziliensis* in skin biopsies of TL carriers in the city of Tomé-Açu [2]. In this same locality, we detected canine infection by *L.* (*V.) braziliensis* and reported the first case of *L**. (V.)*
*infection. guyanensis* in dogs [17].

In this small series of cases, the clinical profile (single ulcerated lesion) and demographics of the patients resembled that in the commonly observed areas with endemic TL [38]. Multiple injuries were identified in three patients, which may have occurred due to infection with *L. (V.) braziliensis*.

Normally, species from the same geographical regions tend to cluster in the same clade [39]. Therefore, the topology of the phylogenetic tree estimated for the *L. (**V.*) *braziliensis* sequences grouped into distinct clades may be the result of possible genetic mutations in the locality. In the Nordeste region of Brazil, there are reports of genetic variability and polymorphisms of the species, resulting in phylogenetic differences, although they present close genetic content [40,41].

## 5. Conclusions

Here, we report the first *L. (V.) shawi* infection in mucosal secretion in Brazil; particularly, from a healthy woman living in the same household with one of the patients with TL participating in the study also infected with *L. (V.) shawi*. This finding represents a warning for the possible association between *L. (V.) shawi* and mucous lesions in the Amazon. We also highlighted the importance of metagenomic studies in investigating *Leishmania* sp. infection by *LRV1* in cases of mucosal leishmaniasis, which could elucidate the pathogenic action of *L. (V.) shawi* and other *Leishmania* species.

Four haplotypes or genetic variants of *L. (V.) braziliensis* based on the gene sequence *hsp70-234* were identified, and the same genetic variability was not detected in *L. (V.) shawi*. The relationship between genetic variants and the severity of TL still needs to be studied, since mutations could influence the evolution and outcome of TL treatment. Clinical studies to establish this relationship are necessary and cannot be dispensed with the molecular characterization of *Leishmania*.

## Figures and Tables

**Figure 1 ijerph-19-06346-f001:**
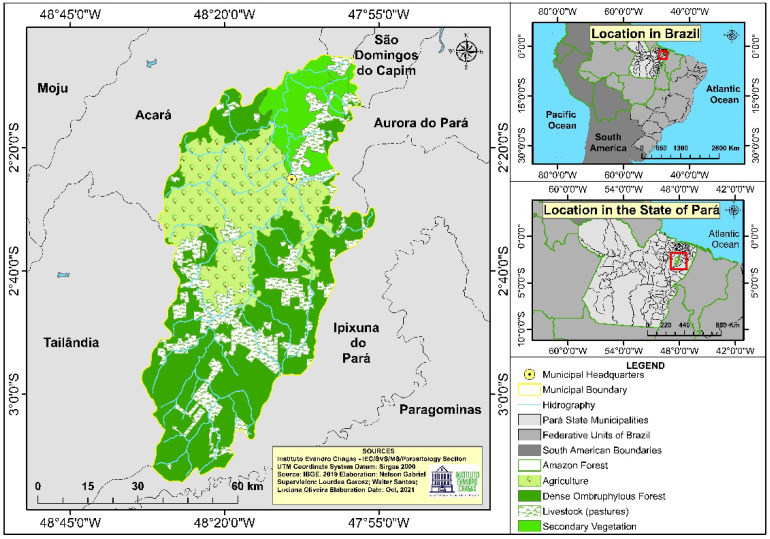
Location and vegetation map of Tomé-Açu, Pará, Amazon, Brazil.

**Figure 2 ijerph-19-06346-f002:**
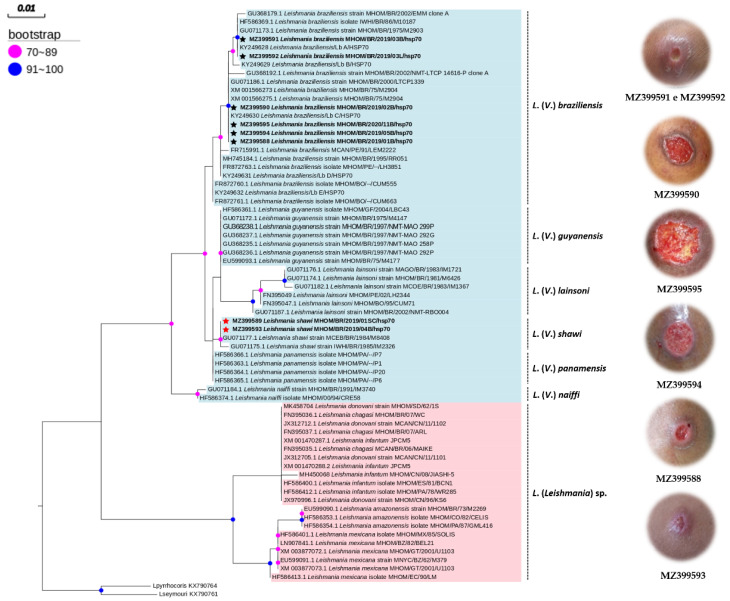
Phylogenetic tree of the genus *Leishmania* estimated using maximum likelihood of *hsp70-234* (234 bp). Sequences of *Leptomonas pyrrhocons* and *Leptomonas seymourii* were used as an external group. The tree involved 67 nucleotide sequences, 7 of which were obtained in this study. The identified sequences of *L. (V.) shawi* are identified with a star (★) and *L. (V.) braziliensis* also marked by a (★), and were related to the clinical characteristics of the lesions observed during the study.

**Table 1 ijerph-19-06346-t001:** TL in an endemic area of the Amazon: etiology of *Leishmania* sp. infection in individuals from the city of Tomé-Açu, in northeastern Pará, Brazil. (+) Amplified DNA; (−) Unamplified DNA; (•) DNA sequenced.

Samples		PCR-*Hsp70-234*			Diagnosis
		Biopsy of the Lesion	*Injury Swab*	Nasal *Swab*	
Patient with TL	1D	+•	+	−	*L. (V.) braziliensis*
	2D	+•	+	+	*L. (V.) braziliensis*
	3D	+•	+•	+	*L. (V.) braziliensis*
	4D	+•	−	−	*L. (V.) shawi*
	5D	+•	−	−	*L. (V.) braziliensis*
	6D	−	+	−	*Leishmania* sp.
	7D	+	−	−	*Leishmania* sp.
	8D	−	−	−	Undetectable DNA
	9D	−	+	+	*Leishmania* sp.
	10D	−	−	−	Undetectable DNA
	11D	+•	+	+	*L. (V.) braziliensis*
Healthy Individuals	1S			+	*Leishmania* sp.
	2S			−	Undetectable DNA
	3S			+	*Leishmania* sp.
	4S			+•	*L. (V.) shawi*
	5S			+	*Leishmania* sp.
	6S			+	*Leishmania* sp.

**Table 2 ijerph-19-06346-t002:** Sociodemographic information of patients with TL (n = 9) and characteristics of lesions in the localized cutaneous form and their etiological agents.

Variables	Species		
	*L.* (*V.*) *braziliensis*	*L.* (*V.*) *shawi*	*Leishmania* sp.
**Sex**			
Male	4	1	3
Female	1	0	0
**Age**			
21–30	3	1	3
31–40	2	0	0
**Number of injuries**			
One site	3	1	3
Multiple	2	0	0
**Kind**			
Ulcerated	5	1	3
**Size (cm)**			
2–6	3	1	0
7–10	1	0	1
11–14	1	0	0
20–24	0	0	1
31–34	0	0	1
**Location**			
Head	1	0	0
Upper limbs	0	1	0
Lower Limbs	4	0	3
**N**	**5**	**1**	**3**

**Table 3 ijerph-19-06346-t003:** Haplotypes of *Leishmania* species for the *hsp70-234*gene.

Species	Haplotype	GenBank Access Number	Nucleotide Position				
			88	122	125	167	191
*L.* (*V.*) *braziliensis*	Hap I	MZ399588 and MZ399590	T	A	G	G	G
	Hap II	MZ399591 and MZ399592	C	G	T	T	A
	Hap III	MZ399594	C	G	T	T	A
	Hap IV	MZ399595	C	G	T	T	R
*L.* (*V.*) *shawi*	Hap V	MZ399593 and MZ399589	C	G	T	T	G

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
