# Peer review of "First Report of an Asymptomatic Leishmania (Viannia) shawi Infection Using a Nasal Swab in Amazon, Brazil"

_ijerph, 2022, doi:10.3390/ijerph19106346_

Round 1
Reviewer 1 Report
Even if the authors can present only a small number of patients, the study presents new insights. All in all, the. manuscript can be recommended for publication, but first it would be helpful, if. the authors could comment on following issue:
- What diagnostic tool can the authors recommend for diagnosis regarding the different sensitivity between nasal swab, lesion swab and biopsy?
Reviewer 2 Report
The authors of the manuscript: “First Report of an Asymptomatic Leishmania (Viannia) shawi Infection using a Nasal Swab in Amazon, Brazil” present a research study where they used PCR and sequencing of an HSP70 gene (a sequence of 234 bp) to detect and discriminate Leishmania species.
The results are remarkable because they first report an asymptomatic Leishmania (Viannia) shawi using a nasal swab. However, the authors do not include the sequencing of another gene or other strategies (like genotyping) to support these findings. Could the authors include another strategy to support the data obtained in the present manuscript?
A list of some things that need to be improved in the manuscript:
1. There are not enough references in the introduction of the manuscript.
2. Improve the grammar. Lane 25 was not written in English.
3. Could the authors indicate the importance of collecting the samples on the indicated dates (stages)? Does it have to do with climate changes, and where are there higher infection rates?
4. There are several words without space in the manuscript (some examples: Lanes 107, 68, 173, 205, and others.)
5. Company names/brands were written in italics and sometimes with the regular font (see paragraphs for methodologies 2.8 and 2.9); please uniformize.
Reviewer 3 Report
Dear Authors,
The manuscript was interesting, as you found that it was the first time that L. (V.) shawi is isolated from a healthy human. However, there are some more changes that should make.
Abstract
In “material and methods” you write that the number of patients was 11 and the number of the control group, 6. In “result” you write that the patients with TL are 9 and the number of the control group is 5. After reading the manuscript, I understand that there was no isolation of RNA from 1 person from the control group and 2 of the patients with TL. This should be written in the abstract, to avoid misunderstanding.
Introduction
I think you should write a few words about Leishmania (Viannia) lainsoni, L. (Viannia) naiffi, L. (Viannia) lindenbergi, and especially for Leishmania (Viannia) shawi, as it is the main element of this manuscript.
L 53-57: You should mention the aim of the study not what you have done. This paragraph should be removed in the “discussion section”.
Material and Methods
L 103-104: why the positive controls were L. infantum and L. (V.) braziliensis? You have not mentioned L. infantum anywhere else in the text.
Results:
Table 1: “TL in an endemic area of the Amazon: etiology of Leishmania sp. infection in individuals from the city of Tomé-Açu, in northeastern Pará, Brazil. The six healthy individuals (1S to 6S) were accompanied by six of the eleven patients (1 to 6). L.(V.) shawi was detected in a nasal swab sample of a healthy individual, confirmed through sequencing of the gene region hsp70-234. This is the same species identified through sequencing of the DNA extracted from the skin of the patient residing in the same house”. The title you have written can be removed to the text. The title of a table should describe to the point what we see in it.
Table 1: why you have used 2 samples for sequencing from the patient 3D, and not from the others (1D, 2D, 11D), and why only from biopsy and not from nasal samples? What were the criteria for the selection of the samples that you sequenced? Why you have not used samples for sequencing, from patients with Leishmania sp.?
Table 2: “For discrimination of Leishmania species, amplification and sequencing of the gene region hsp70-234 of Leishmania was performed.” This can be removed as a footnote.
L 169: 57 nucleotide sequences? 7 sequences in this study?
L 223: “patient(skin” please insert space between “patient” and “(skin”.
Round 2
Reviewer 2 Report
Dear researchers, I appreciate that you have completed some changes, however. I have not been able to find a bibliographical reference that indicates the amplification of a fragment of the HSP70 gene and sequencing is sufficient to identify a species.
If you have the referenced manuscript, I would ask you to please cite it.
Otherwise, amplification and sequencing of another gene would be necessary, or some typing assay.
Author Response
Dear Reviewer, we added the reference on the use of the marker for species identification, and published articles that identified species with the sequencing of the region.
Lane: 63-70